# SINS/CNS/GNSS Integrated Navigation Based on an Improved Federated Sage–Husa Adaptive Filter

**DOI:** 10.3390/s19173812

**Published:** 2019-09-03

**Authors:** Shuqing Xu, Haiyin Zhou, Jiongqi Wang, Zhangming He, Dayi Wang

**Affiliations:** 1College of Liberal Arts and Sciences, National University of Defense Technology, Changsha 410073, China; 2Beijing Institute of Spacecraft System Engineering, China Academy of Space Technology, Beijing 100094, China

**Keywords:** multi-source navigation, federated filter, Sage–Husa adaptive filter, time-varying state noise, biased estimation, weighting function, exponential function

## Abstract

Among the methods of the multi-source navigation filter, as a distributed method, the federated filter has a small calculation amount with Gaussian state noise, and it is easy to achieve global optimization. However, when the state noise is time-varying or its initial estimation is not accurate, there will be a big difference with the true value in the result of the federated filter. For the systems with time-varying noise, adaptive filter is widely used for its remarkable advantages. Therefore, this paper proposes a federated Sage–Husa adaptive filter for multi-source navigation systems with time-varying or mis-estimated state noise. Because both the federated and the adaptive principles are different in updating the covariance of the state noise, it is required to weight the two updating methods to obtain a combined method with stability and adaptability. In addition, according to the characteristics of the system, the weighting coefficient is formed by the exponential function. This federated adaptive filter is applied to the SINS/CNS/GNSS integrated navigation, and the simulation results show that this method is effective.

## 1. Introduction

With the advancement of the navigation and the technology of information fusion, the multi-source navigation [1] has become the main composition of the integrated navigation with high precision and reliability. In practical applications, due to the geographical location, equipment failure and radio interfered, some navigation modes will not work, but other undisturbed navigation modes will continue to operate, enabling the multi-source navigation to continue navigating for a long time. Through the detection [2] and correlation [3] of the data, information fusion can improve the accuracy of state estimation. In addition, in the field of navigation, the information fusion technology can be used to solve the problem of the low accuracy of a single navigation source in the multi-source navigation [4]. Therefore, the information fusion technology of multi-source navigation is the key to navigation operations.

For the problem of multi-source information fusion, Carlson proposed the federated filter, which can use the information distribution principle to eliminate the correlation of each sub-state estimation. The distributed principle makes the calculation smaller and more fault-tolerant, and global optimal or sub-optimal estimates can be obtained through effective fusion, which makes the federated filter widely used [5].

The federated filter can be composed of one main filter and several local filters, the main filter and the local filters have the same state equation, and the measurement equations of the local filters differ according to the measurement information. In the traditional federated Kalman filter algorithm, due to the federal allocation and weighting principle, the estimated value of the system state noise covariance remains unchanged, that is, the same as the initial estimation. When the estimation of state noise is accurate and there is no time-varying feature, the result of the federated filter is good enough; however, when the initial estimation is not accurate or the state noise is time-varying, there will be a large error in the global filter estimation, which will reduce the navigation accuracy.

For systems with time-varying or mis-estimated state noise, the Sage–Husa adaptive filter [6] uses the time-varying noise statistical estimator to correct the system state noise and observation noise, and the simplified Sage–Husa adaptive filter estimates the current state noise to obtain the adaptive filter value under this estimation by using the forgetting factors [7]; through this process, the model error can be reduced, the filter divergence can be suppressed, and the navigation accuracy can be improved.

In this paper, for multi-source navigation with time-varying or mis-estimated state noise, the federated adaptive Kalman filter is used for the operation, that is, the local filters adopt the simplified Sage–Husa adaptive algorithm [8], while, in the overall framework, the federated principle is used for calculations. In view of the fact that both the federated and the adaptive algorithm have updating principles of the state noise covariance, this paper proposes a weighting method to fuse these two principes. As a result, the federated adaptive updating process can adjust the state noise covariance according to the information distribution factor, so that the global filter can maintain stability on the basis of the adaptive changes. The trend of the weighting value can be determined by analyzing the variation characteristics of the system, and the exponential function is selected to fit the system. Compared with the federated Kalman filter and the common federated adaptive Kalman filter in the simulations, it is found that the improved federated adaptive filter is better in position and speed determination, which verifies the effectiveness of the proposed method.

The contributions of this paper are as follows: the (1) SINS (strapdown inertial navigation system)/CNS (celestial navigation system)/GNSS (global navigation satellite system) integrated navigation mode is established based on the measurement data of various sensors, and federated principle is used for distributed computing; (2) for the case with time-varying or mis-estimated state noise, federated Sage–Husa adaptive filter is chosen as the sub-filter’s algorithm, and the state noise covariance is weighted according to the federated and the adaptive principle to ensure the adaptability and stability of the global filter; (3) In the simulation part, the comparison of the navigation accuracy among federated filter, federated adaptive filter and the improved federated adaptive filter is completed.

The following chapters are structured as follows: Section 2 introduces the principle of federated Kalman filter, including the structure and operation principle of it, as well as the algorithm of the filter. Section 3 introduces the Sage–Husa adaptive Kalman filter algorithm. Section 4 introduces the principle of the improved federated adaptive filter, including the selection process of the federated distribution factors and the adaptive weighting of the state noise covariance. Section 5 constructs the SINS/CNS/GNSS integrated navigation model and gives the state equations and measurement equations of the integrated navigation; Section 6 demonstrates the effectiveness of the improved federated adaptive filter through the simulations; Section 7 is the conclusions.

## 2. Introduction of the Federated Kalman Filter

When the navigation process involves three or more navigation methods, it is difficult to combine the measurement information of each method effectively by using a single filter. For this situation, the researchers have proposed a number of distributed filter methods. The standard distributed algorithm [9] was proposed, which is intended to establish the relationship between the distributed and centralized filter; considering the unknown correlation of local estimations, there is the covariance crossover algorithm [10] as well as the federated algorithm [11].

Federated Kalman filter is a special form of distributed Kalman filter and it was proposed by Carlson in the United States in 1998. It consists of several local filters and one main filter, and it is a decentralized filter method with block estimation and a two-step cascade. It assigns dynamic state and observation information to each local filter and each local filter operates separately. The results of local filters are combined according to the information distribution factors to obtain the result of the global filter. Obviously, the key of the operation lies in the information distribution process.

### 2.1. Principle and Structure of the Federated Kalman Filter

The federated filter operation process utilizes the measurement information of each subsystem and the common reference system for parallel independent operations. Suppose that there are *N* local filters, the subscript of the main filter is *m*, and the subscript of the global filter is *g*, the state and measurement equations of each local filter and the main filter are as follows:(1)$$\left\{\begin{array}{c} X_{i,k}\;=\;\Phi_{i,k-1}X_{i,k-1}+W_{i,k-1}\\ Z_{i,k}\;=\;H_{i,k}X_{i,k}+V_{i,k} \end{array}\right.,i\;=\;1,2,\cdots ,N,\left(m\right),$$
where $X_{i,k}$ is the state vector of the local filter or main filter, $Z_{i,k}$ is the measurement vector, $\Phi_{i,k-1}$ is the state transition matrix of the $i^{th}$ local filter at time k−1; $H_{i,k-1}$ is the measurement matrix; $W_{i,k-1}$ and $V_{i,k}$ are the state noise matrix and measurement noise matrix of the local filter respectively, and they are all Gaussian white noise matrices, the variances are $Q_{i,k-1}$ and $R_{i,k}$ respectively. It should be noted that the main filter has no measurement equation, i.e., when i=m, only the state equation works.

Suppose that the local optimal estimation ${\hat{X}}_{i,k-1}$ and its corresponding covariance $P_{i,k-1}$ are obtained at time k−1, and these local optimal estimations are fused in the global filter according to the optimal fusion estimation algorithm to obtain the global optimal estimation ${\hat{X}}_{g,k-1}$ and its variance $P_{g,k-1}$. The state noise covariance matrices of the local filter and the global filter are $Q_{i,k-1}$ and $Q_{g,k-1}$ respectively, and $P_{g,k-1}$ and $Q_{g,k-1}$ are amplified by $\beta_{i}^{-1}$ times and then fed back to the local filters for parameter reset, i.e., the parameter value of *k* time is obtained:(2)$$\left\{\begin{array}{c} {\hat{X}}_{i,k}\;=\;{\hat{X}}_{g,k-1}\\ P_{i,k}\;=\;\beta_{i}^{-1}P_{g,k-1}\\ Q_{i,k}\;=\;\beta_{i}^{-1}Q_{g,k-1} \end{array}\right.,i\;=\;1,2,\cdots ,N,\left(m\right),$$
where $\beta_{i}$ is the information distribution factor. In addition, according to the principle of information conservation, the information distribution factor $\beta_{i}$ needs to satisfy:(3)$$\left\{\begin{array}{c} \beta_{1}+\beta_{2}+\cdots +\beta_{N}+\left(\beta_{m}\right)\;=\;1,\\ 0\leq \beta_{i}\leq 1. \end{array}\right.$$

At the same time, the federated filter has the following principles of information distribution:(4)$$\left\{\begin{array}{c} Q_{g,k}^{-1}\;=\;Q_{1,k}^{-1}+Q_{2,k}^{-1}+\cdots +Q_{N,k}^{-1}+\left(Q_{m,k}^{-1}\right),\\ P_{g,k}^{-1}\;=\;P_{1,k}^{-1}+P_{2,k}^{-1}+\cdots +P_{N,k}^{-1}+\left(P_{m,k}^{-1}\right). \end{array}\right.$$

Through the above equations, the federated filter links each local filter with the main filter, and realizes the fusion process through information distribution, and different federated modes can be obtained by setting different information distribution factor $\beta_{i}$ [12]. The improved federated filter reset method proposed in this paper uses Equations (Equation 2) and (Equation 4) to complete the information fusion process through the allocation and addition of global filter and local filter without the participation of the main filter.

For the integrated navigation of SINS, CNS and GNSS in this paper, two local filters are set—SINS/CNS local filter 1 and SINS/GNSS local filter 2, each of which is independent in data processing. As for the setting of the main filter, it is necessary to consider the actuality of the system. For this system, in the case that the initial state noise estimation is not accurate or the state noise is time-varying, the main filter is not accurate without the measurement equation, so the main filter can be left. The data of each navigation subsystem is input to the corresponding local filter, and the output is the result of information fusion, and the global filter result can be obtained, then the global state estimation is realized. The structure of the federated filter is as Figure 1:

As can be seen from Figure 1, on the one hand, the information from the global filter is output to the outside, and, on the other hand, it is fed back to each sub-filter. The existence of the feedback process makes the information fusion process of the distributed filter more efficient and accurate.

### 2.2. Algorithm Flow of the Federated Kalman Filter

For the federated filter structure without the main filter (i.e., $\beta_{1}+\beta_{2}+\cdots +\beta_{N}\;=\;1$), parameters and their changes of the local filter affect the result of the global filter [13]. Taking the discrete model in Equation (Equation 1) as an example, the steps of the federated filter algorithm are mainly as follows:

a. Initialization:

Firstly, global estimation initialization is performed, and the initial value of the state vector ${\hat{X}}_{g,0}$, the initial value of the state covariance $P_{g,0}$, and the initial value of the state noise $Q_{g,0}$ are known.

b. Information distribution (reset):

Secondly, the information distribution process is as follows:(5)$$P_{i,k}\;=\;\beta_{i}^{-1}P_{g,k-1},$$
(6)$$Q_{i,k}\;=\;\beta_{i}^{-1}Q_{g,k-1},$$
(7)$${\hat{X}}_{i,k}\;=\;{\hat{X}}_{g,k-1}.$$

In this process, the value of $\beta_{i}$ affects the proportion of each local filter, and the principles of subsystems are not the same as each other. The specific selection principle is described in Section 4.1:

c. Local estimation:

The state prediction:(8)$${\hat{X}}_{i,k|k-1}\;=\;\Phi_{k-1}{\hat{X}}_{i,k-1},$$

The variance prediction:(9)$$P_{i,k|k-1}\;=\;\Phi_{k-1}P_{i,k-1}\Phi_{k-1}^{T}+Q_{i,k-1},$$

The variance is updated:(10)$$P_{i,k}\;=\;\left(I-K_{i,k}H_{i,k}\right)P_{i,k|k-1},$$

The state measurement is updated:(11)$$\begin{array}{c} {\hat{X}}_{i,k}\;=\;{\hat{X}}_{i,k|k-1}+K_{i,k}\left(Z_{i,k}-H_{i,k}{\hat{X}}_{i,k|k-1}\right),\\ K_{i,k}\;=\;P_{i,k|k-1}H_{i,k}{\left(H_{i,k}P_{i,k|k-1}H_{i,k}^{T}+R_{i,k}\right)}^{-1}, \end{array}$$

d. Global integration:

The variance fusion:(12)$$P_{g,k}\;=\;{\left[{\sum_{i\;=\;1}^{N}\left(P_{i,k}\right)}^{-1}\right]}^{-1},$$

The state fusion:(13)$${\hat{X}}_{g,k}\;=\;P_{g,k}\left[\sum_{i\;=\;1}^{N}{\left(P_{i,k}\right)}^{-1}{\hat{X}}_{i,k}\right].$$

After each round of the filter calculation process, it will return to the information distribution (reset) link to start the next round of calculation.

## 3. Introduction of the Sage–Husa Adaptive Filter

The Sage–Husa algorithm is an adaptive filter algorithm based on the statistical characteristics of the system [14]. For the case that the statistical properties of the state and measurement noise are unknown, the maximal posterior estimation principle can be used to obtain the estimated value [15] to improve the filter accuracy. The estimation algorithm is suitable for general linear time-varying systems. The recursive calculation process is simple and suitable for many fields.

Consider the mathematical model of the linear discrete systems:(14)$$\left\{\begin{array}{c} X_{k}\;=\;\Phi_{k-1}X_{k-1}+W_{k-1},\\ Z_{k}\;=\;H_{k}X_{k}+V_{k}, \end{array}\right.$$
where $\Phi_{k-1}$ is the state transition matrix; $H_{k-1}$ is the measurement matrix; $W_{k-1}$ and $V_{k}$ are the state noise matrix and the measurement noise matrix, and the covariance matrices are $Q_{k-1}$ and $R_{k}$, respectively, and their statistical properties are unknown.

For the systems where the variance $W_{k}$ of measurement noise is time-varying or unknown, the general Kalman filter algorithm is difficult to meet the accuracy requirements of the system due to the lack of updating procedures for the system and measurement noise. From the aspect of optimizing the filter performance, the contribution rate of the new data to the filter can be correspondingly improved, so the operator $d_{k}$ is needed, satisfying
(15)$$d_{k}\;=\;\frac{1-b}{1-b^{k+1}},$$
where *b* is the forgetting factor, and 0<b<1. The corresponding iterative factor’s updating process is as follows:(16)$${\hat{q}}_{k}\;=\;\left(1-d_{k-1}\right){\hat{q}}_{k-1}+d_{k-1}\left[{\hat{X}}_{k}-\Phi_{k}{\hat{X}}_{k-1}\right],$$
(17)$${\hat{Q}}_{k}\;=\;\left(1-d_{k-1}\right){\hat{Q}}_{k-1}+d_{k-1}\left[K_{k}{\tilde{Z}}_{k}{\tilde{Z}}_{k}^{T}K_{k}^{T}+P_{k}-\Phi_{k-1}P_{k-1}\Phi_{k-1}^{T}\right],$$
(18)$${\hat{r}}_{k}\;=\;\left(1-d_{k-1}\right){\hat{r}}_{k-1}+d_{k-1}\left[Z_{k}-H_{k}{\hat{X}}_{k|k-1}\right],$$
(19)$${\hat{R}}_{k}\;=\;\left(1-d_{k-1}\right){\hat{R}}_{k-1}+d_{k-1}\left[{\tilde{Z}}_{k}{\tilde{Z}}_{k}^{T}-H_{k}P_{k|k-1}H_{k}^{T}\right],$$
where ${\hat{q}}_{k}$ and ${\hat{r}}_{k}$ are the estimates of the mathematical expectation of the system error and measurement error at time *k*, respectively. ${\hat{Q}}_{k}$ and ${\hat{R}}_{k}$ are the estimates of the variance of the system error and measurement error at time *k*, respectively. Combining the above iterative factors with the Kalman filter algorithm, a robust adaptive Kalman filter algorithm which can automatically track noise can be obtained as follows:

The one-step prediction equation:(20)$${\hat{X}}_{k|k-1}\;=\;\Phi_{k-1}{\hat{X}}_{k-1}+{\hat{q}}_{k},$$

The mean square error of the one-step prediction:(21)$$P_{k|k-1}\;=\;\Phi_{k-1}P_{k-1}\Phi_{k-1}^{T}+{\hat{Q}}_{k-1},$$

The gain of the filter:(22)$$K_{k}\;=\;P_{k|k-1}H_{k}^{T}{\left[H_{k}P_{k|k-1}H_{k}^{T}+{\hat{R}}_{k}\right]}^{-1},$$

The estimation of the mean square error:(23)$$P_{k}\;=\;\left[I-K_{k}H_{k}\right]P_{k|k-1},$$

The state estimation:(24)$${\hat{X}}_{k}\;=\;{\hat{X}}_{k|k-1}+K_{k}{\tilde{Z}}_{k}.$$

By adjusting the forgetting factor *b*, the adaptive process of the system can be fulfilled.

## 4. Improved Federated Adaptive Filter Algorithm

### 4.1. Selection of the Federated Filter Information Distribution Factors

It is known that the structure and parameter updating process of federated filter is closely related to the selection of information distribution factor $\beta_{i}$ [16]. Therefore, it is necessary to select the appropriate $\beta_{i}$ according to the characteristics of the system to achieve better filter effect.

In the present literature, the selection methods of $\beta_{i}$ are mainly divided into two types, one is based on the fixed ratio [17], which is suitable for the process without dynamic changes or the proportion of state covariance remains unchanged. For example, when the parameters of each local filter are the same, the distribution can be set as $\beta_{i}\;=\;\frac{1}{N}$. The other method is used for the case in which the relevant parameters of the subsystem change with time. In this time, the dynamic adaptive method can be used to select the information distribution factor [18]. The distribution methods are mainly divided into several types:

(1) According to the trace of the $P_{i}$ matrix [19,20]:

Let
(25)$$\beta_{i}\;=\;\frac{trP_{i}}{\sum_{i\;=\;1}^{N,m}trP_{i}}.$$

The information distribution factor can be obtained by estimating the state vector covariance matrix $P_{i}$.

(2) According to the F norm of the P matrix [21]:(26)$$\beta_{i}\;=\;\frac{{∥P_{i,k-1}∥}_{F}}{\sum_{i\;=\;1}^{n}{∥P_{i,k-1}∥}_{F}}\left(1-\beta_{m}\right).$$

Since the parameters of the local filters are not the same and it cannot guarantee that the parameter weight remains unchanged, it is necessary to select an information distribution factor with dynamic adaptive ability. Considering the computational complexity of these algorithms, this paper chooses Equation (Equation 25) as the solution algorithm of $\beta_{i}$.

### 4.2. Selection of Federated Adaptive Filter’s Partition Coefficient and Its Feasibility Analysis

#### 4.2.1. Selection of Federated Adaptive Filter’s Partition Coefficient and Its Feasibility Analysis

In the actual situations, the statistical properties of the state noise are often difficult to determine, and the inaccurate state noise covariance will affect the accuracy of the filter. Therefore, in the framework of the federated filter, the simplified Sage–Husa adaptive filter [22] can be chosen as the algorithm of the local filter, thus an improved federated adaptive filter algorithm can be proposed.

The traditional federated Kalman filter does not have the ability to eliminate the influence of deviation. For the state noise covariance, after that, the initial value $Q_{g,0}$ is given, the iterative process at each moment simply re-updates the value of $Q_{g,0}$ according to the information distribution factor. When there is a deviation in the initial value, the deviation will always exist in the filter process, which will affect the filter result. Assume that
(27)$$Q_{0}\;=\;\Delta Q_{0}+Q_{g,0},$$
where $Q_{0}$ is the true value of the initial state noise, $\Delta Q_{0}$ is the deviation between the true value and the estimated value. Due to the existence of $\Delta Q_{0}$, the filter effect of the traditional federated Kalman filter is difficult to guarantee.

When the Sage–Husa adaptive filter is selected by local filter, the influence of the initial deviation on the filter is gradually weakened due to the update of ${\hat{Q}}_{i,k}$, which makes the filter more adaptable.

In fact, the measurement noise of the system is related to the accuracy of the measuring instrument, the distance and the angle of the target. In this paper, it is assumed that the statistical properties of the measurement noise are known, and the simplified Sage–Husa adaptive algorithm can be obtained by using statistical characteristics of state noise [23].

During the operation of federated adaptive filter, the iterative process of federated filter continuously updates ${\hat{X}}_{g,k}$, $P_{g,k}$, and $Q_{g,k}$ through Equations (Equation 2) and (Equation 4), while adaptive filter updates ${\hat{q}}_{i,k}$ and ${\hat{Q}}_{i,k}$ through Equations (Equation 16) and (Equation 17). Since there may be a deviation in the initial value of the state noise covariance, it is considered to combine the federated updating principle with the adaptive principle, and use the combined federated adaptive principle to update the covariance of the state noise.

For each local filter, it is assumed that there are two updating methods—the federated principle and the adaptive principle method, which are as follows:(28)$${\hat{Q}}_{i,k+1}^{1}\;=\;\beta_{i}^{-1}{\hat{Q}}_{g,k},$$
(29)$${\hat{Q}}_{i,k+1}^{2}\;=\;\left(1-d_{k}\right){\hat{Q}}_{k}+d_{k}\left[K_{k+1}{\tilde{Z}}_{k+1}{\tilde{Z}}_{k+1}^{T}K_{k+1}^{T}+P_{k+1}-\Phi_{k}P_{k}\Phi_{k}^{T}\right],$$
where ${\hat{Q}}_{i,k+1}^{1}$ and ${\hat{Q}}_{i,k+1}^{2}$ are the state noise covariance estimations of the *i*th filter at k+1 moment by using the federated algorithm and the adaptive algorithm, respectively. It is known that the updating process of the federated principle is related to the initial value. When the initial value is accurate or it is Gaussian white noise, it can use the information distribution factor to obtain the optimal solution globally; in addition, for the system with inaccurate or time-varying value, the adaptive updating process can adjust the adaptive degree of the filter by selecting the operator $d_{k}$ [24], and it is related to the forgetting factor *b*.

In the operation of improved federated adaptive filter, the proportion of adaptive algorithm distribution increases with the change of state noise. Consider weighting the two update processes to get the following equation:(30)$${\hat{Q}}_{i,k+1}\;=\;\alpha {\hat{Q}}_{i,k+1}^{1}+\left(1-\alpha \right){\hat{Q}}_{i,k+1}^{2}.$$

According to the variation characteristics of the state noise, the proportion of α in the equation should decrease, and the federated adaptive filter should always satisfy 0<α<1. In the first quadrant, the changes of the linear function do not satisfy the above conditions, and the inverse proportional function, the transformed exponential function and logarithmic function can satisfy the conditions. In this paper, the transformed exponential function is selected as the changing function of the weight, that is,
(31)$$\alpha_{k}\;=\;\frac{\sigma }{e^{k}},$$
where $\alpha_{k}$ is the weighting ratio of the federated method at *k* time; σ>0, σ is chosen to control the rate of the change of α.

The mean square error (MSE) of state noise satisfies
(32)$$\mathrm{MSE}\left(Q\right)\;=\;\mathrm{bias}\left(Q\right)+\mathrm{var}\left(Q\right),$$
where $\mathrm{bias}\left(Q\right)$ is the deviation of state noise, $\mathrm{var}\left(Q\right)$ is the variance. There will be a deviation in the setting of the initial value according to the federated principle, and the result of the adaptive filter will have a large variance when the number of samples is small. Therefore, the deviation of the state noise variance is mainly from the federated updating method, and the variance mainly comes from the adaptive updating method. For the sake of convenience, according to the variation characteristics of the weight, the initial variance of ${\hat{Q}}_{i,k+1}^{1}$ in the federated algorithm is set to 0, and the initial deviation of ${\hat{Q}}_{i,k+1}^{2}$ in the adaptive algorithm is set to 0. Thus, the mean square error of the state noise variance estimation of the federated adaptive filter at k+1 time should satisfy the following equation:(33)$$\begin{array}{c} \mathrm{MSE}\left({\hat{Q}}_{i,k+1}\right)\;=\;\alpha^{2}\mathrm{MSE}\left({\hat{Q}}_{i,k+1}^{1}\right)+{\left(1-\alpha \right)}^{2}\mathrm{MSE}\left({\hat{Q}}_{i,k+1}^{2}\right)\\ \;=\;\alpha^{2}\left(\mathrm{bias}\left({\hat{Q}}_{i,k+1}^{1}\right)+\mathrm{var}\left({\hat{Q}}_{i,k+1}^{1}\right)\right)+{\left(1-\alpha \right)}^{2}\left(\mathrm{bias}\left({\hat{Q}}_{i,k+1}^{2}\right)+\mathrm{var}\left({\hat{Q}}_{i,k+1}^{2}\right)\right). \end{array}$$

After analysis, it can be seen that $\mathrm{bias}\left({\hat{Q}}_{i,k+1}^{1}\right)$ remains unchanged and it exists at the initial time,$\mathrm{var}\left({\hat{Q}}_{i,k+1}^{1}\right)\;=\;0$; while $\mathrm{var}\left({\hat{Q}}_{i,k+1}^{2}\right)$ has a large value in the initial time due to the few samples, and it gradually decreases with the number of the samples increases, and $\mathrm{bias}\left({\hat{Q}}_{i,k+1}^{2}\right)\;=\;0$. Thus, Equation (Equation 33) can be changed as:(34)$$\mathrm{MSE}\left({\hat{Q}}_{i,k+1}\right)\;=\;\alpha^{2}\mathrm{bias}\left({\hat{Q}}_{i,k+1}^{1}\right)+{\left(1-\alpha \right)}^{2}\mathrm{var}\left({\hat{Q}}_{i,k+1}^{2}\right).$$

#### 4.2.2. Feasibility Analysis of the Federated Adaptive Filter’s Partition Coefficient

According to Equation (Equation 34), in the updating process of ${\hat{Q}}_{i,k+1}$ by the federated adaptive algorithm, $\mathrm{MSE}\left({\hat{Q}}_{i,k+1}\right)$ consists of two parts, and $\mathrm{bias}\left({\hat{Q}}_{i,k+1}^{1}\right)$ remains invariant after the initial value is determined. Therefore, it is necessary to ensure that $\mathrm{var}\left({\hat{Q}}_{i,k+1}^{2}\right)$ decreases with time, thus the feasibility and superiority of the algorithm are guaranteed.

For Equation (Equation 34), assume that $\Omega_{k}\;=\;K_{k+1}{\tilde{Z}}_{k+1}{\tilde{Z}}_{k+1}^{T}K_{k+1}^{T}+P_{k+1}-\Phi_{k}P_{k}\Phi_{k}^{T}$,let $\mathrm{var}\left(\Omega_{k}\right)\;=\;\Delta_{k}$, then
(35)$$\begin{array}{c} \mathrm{var}\left({\hat{Q}}_{i,k}^{2}\right)\;=\;\Delta_{2}\prod_{i\;=\;2}^{k-1}{\left(1-d_{i}\right)}^{2}+\sum_{i\;=\;3}^{k-1}{d_{i-1}^{2}\Delta_{i}\prod_{j\;=\;3}^{i+1}\left(1-d_{j}\right)}^{2}+d_{k-1}^{2}\Delta_{k}\\ \;=\;\Delta_{2}\prod_{i\;=\;2}^{k-1}{\left(1-d_{i}\right)}^{2}+\sum_{i\;=\;3}^{k-1}\left[d_{i-1}^{2}\Delta_{i}\prod_{j\;=\;3}^{i}{\left(1-d_{j}\right)}^{2}\right]+d_{k-1}^{2}\Delta_{k} \end{array}$$
(36)$$\begin{array}{c} \mathrm{var}\left({\hat{Q}}_{i,k+1}^{2}\right)\;=\;{\left(1-d_{k}\right)}^{2}\mathrm{var}\left({\hat{Q}}_{i,k}^{2}\right)+d_{k}^{2}\Delta_{k+1}\\ \;=\;\mathrm{var}\left({\hat{Q}}_{i,k}^{2}\right)-2d_{k}\mathrm{var}\left({\hat{Q}}_{i,k}^{2}\right)+d_{k}^{2}\left(\mathrm{var}\left({\hat{Q}}_{i,k}^{2}\right)+\Delta_{k+1}\right). \end{array}$$

To make $\mathrm{var}\left({\hat{Q}}_{i,k+1}^{2}\right)<\mathrm{var}\left({\hat{Q}}_{i,k}^{2}\right)$, then
(37)$$-2d_{k}\mathrm{var}\left({\hat{Q}}_{i,k}^{2}\right)+d_{k}^{2}\left(\mathrm{var}\left({\hat{Q}}_{i,k}^{2}\right)+\Delta_{k+1}\right)<0.$$

Simplified:(38)$$d_{k}\Delta_{k+1}<\left(2-d_{k}\right)\mathrm{var}\left({\hat{Q}}_{i,k}^{2}\right).$$

It can be seen from Equation (Equation 15) that the operator $d_{k}$ can be controlled by selecting the forgetting factor *b*, so the federated adaptive algorithm is feasible under the conditions of Equation (Equation 38).

The Sage–Husa adaptive filter has a small sample size at the initial time, and the estimated state noise variance has a large variance. At this time, if the value of the forgetting factor *b* is increased, the adaptive convergence will slow down. Therefore, the integrated method can guarantee the convergence speed as well as the estimation accuracy. The dynamic information distribution of federated adaptive filter is completed by using the exponential function as the weighting algorithm.

In summary, it is assumed that ${\hat{Q}}_{k}$ is the state noise variance estimation at *k* time of the federated adaptive algorithm, the algorithm flow of the federated adaptive filter is as follows:

Through the operation flow shown in Figure 2, a federated adaptive algorithm can be obtained, which is adaptive and stable to meet the requirements of the multi-source system navigation with unknown state noise characteristics.

## 5. SINS/CNS/GNSS Integrated Navigation Model

ENU geography coordinate system(t): The origin of the coordinate system is the center of the carrier, the $x_{t}$ axis points eastward along the direction of the reference ellipsoid ring, the $y_{t}$ axis points north along the direction of the reference ellipsoid meridian, and the $z_{t}$ axis is determined by the right-hand rule.Aircraft body coordinate system(b): Taking the satellite as an example, the body coordinate system is a coordinate system fixed on the satellite body. The coordinate origin is the satellite centroid, and the $x_{b}$ axis, $y_{b}$ axis and $z_{b}$ axis are usually defined on the satellite’s inertia main axis.Navigation coordinate system(n): The navigation coordinate system is the coordinate system selected according to the needs of solving the navigation parameters.

This paper selects SINS, CNS and GNSS as the three basic navigation methods. By using the high-precision attitude information provided by CNS and the position as well as the velocity information provided by GNSS, the local filters use the Sage–Husa adaptive filter to estimate the position, velocity and attitude errors of SINS accurately, and correct the inertial device error of the SINS. Finally, the system will achieve continuous high-precision navigation of the aircraft.

As shown in Figure 1, in this paper, there is no main filter; two local filters are used to implement the federated filter. They are SINS/CNS local filter 1 and SINS/GNSS local filter 2, respectively. The ENU coordinate system is used as the reference coordinate system, the flight height is assumed as *h*, and the earth is assumed as a spheroid.

### 5.1. The State Equation of the Integrated Navigation System

The state equation of the SINS/CNS/GNSS integrated navigation system consists of the error equations of SINS and the inertial devices, in the form of
(39)$$X_{k}\;=\;\Phi_{k-1}X_{k-1}+G_{k-1}W_{k-1}.$$

Take the state parameter of the system as 15 dimensions, and record it as:(40)$$X\;=\;{\left[\begin{array}{cccccccc} \phi_{E} & \phi_{N} & \phi_{U} & \delta v_{E} & \delta v_{N} & \delta v_{U} & \delta L & \delta \lambda \\ \delta h & \varepsilon_{x} & \varepsilon_{y} & \varepsilon_{z} & \nabla_{x} & \nabla_{y} & \nabla_{z} & \end{array}\right]}^{T},$$
where $\left[\begin{array}{ccc} \phi_{E} & \phi_{N} & \phi_{U} \end{array}\right]$ denotes the three mathematical platform angles error; $\left[\begin{array}{ccc} \delta v_{E} & \delta v_{N} & \delta v_{U} \end{array}\right]$ denotes the velocity error on three axes; $\left[\begin{array}{ccc} \delta L & \delta \lambda & \delta h \end{array}\right]$ denotes the latitude, longitude and height error; $\left[\begin{array}{ccc} \varepsilon_{x} & \varepsilon_{y} & \varepsilon_{z} \end{array}\right]$ and $\left[\begin{array}{ccc} \nabla_{x} & \nabla_{y} & \nabla_{z} \end{array}\right]$ are the gyro random constant drift and the accelerometer random constant drift.

The state noise consists of the random error of the gyroscope and the accelerometer. The expression is
(41)$$W\;=\;{\left[\begin{array}{cccccc} w_{\varepsilon_{x}} & w_{\varepsilon_{y}} & w_{\varepsilon_{z}} & w_{\nabla_{x}} & w_{\nabla_{y}} & w_{\nabla_{z}} \end{array}\right]}^{T}.$$

State noise transformation matrix is:(42)$$G\;=\;\left[\begin{array}{cc} C_{b}^{n} & 0_{3\times 3}\\ 0_{3\times 3} & C_{b}^{n}\\ 0_{9\times 3} & 0_{9\times 3} \end{array}\right],$$
where $C_{b}^{n}$ denotes the rotation matrix of the aircraft body coordinate system to the navigation coordinate system.

### 5.2. The Measurement Equation of the Integrated Navigation System

It is known that the federated adaptive filter of the integrated navigation system contains two local filters, and the ENU geography coordinate system is selected as the navigation coordinate system.

The SINS/CNS subsystem uses the transformed mathematical platform angles error as the measurement vector of the Sage–Husa adaptive filter. The measurement equation is
(43)$$Z_{1,k}\;=\;H_{1,k}X_{k}+V_{1,k},$$
where $Z_{1,k}$ denotes the measurement vector, $Z_{1}\;=\;{\left[\begin{array}{ccc} \phi_{E} & \phi_{N} & \phi_{U} \end{array}\right]}^{T}$; $H_{1,k}$ denotes the measurement matrix, $H_{1}\;=\;{\left[\begin{array}{cc} I_{3\times 3} & 0_{3\times 12} \end{array}\right]}^{T}$. $V_{1}\;=\;\left[\begin{array}{c} \delta \Delta_{x}\\ \delta \Delta_{y}\\ \delta \Delta_{z} \end{array}\right]$, $V_{1}$ denotes the difference between the star sensor and the gyroscope drift error.

The SINS/GNSS subsystem uses the difference between the position and velocity of SINS and GNSS as the measurement information of adaptive filter. The measurement equation is
(44)$$Z_{2,k}\;=\;H_{2,k}X_{k}+V_{2,k}\;=\;\left[\begin{array}{c} H_{2,k}^{v}\\ H_{2,k}^{p} \end{array}\right]X_{k}+\left[\begin{array}{c} V_{2,k}^{v}\\ V_{2,k}^{p} \end{array}\right],$$
where $H_{2}^{v}\;=\;\left[\begin{array}{ccc} 0_{3\times 3} & diag\left(\begin{array}{ccc} 1 & 1 & 1 \end{array}\right) & 0_{3\times 9} \end{array}\right]$,$H_{2}^{p}\;=\;\left[\begin{array}{ccc} 0_{3\times 6} & diag\left(\begin{array}{ccc} R_{M} & R_{N}\cos L & 1 \end{array}\right) & 0_{3\times 6} \end{array}\right]$. $V_{2}^{v}\;=\;\left[\begin{array}{c} v_{E}\\ v_{N}\\ v_{U} \end{array}\right]$, $V_{2}^{v}$ denotes the speed difference between the SINS and GNSS in the three directions; $V_{2}^{p}\;=\;\left[\begin{array}{c} p_{E}\\ p_{N}\\ p_{U} \end{array}\right]$, $V_{2}^{p}$ denotes the position difference between the SINS and GNSS in the three directions.

## 6. Simulation and Analysis

Assume that the trajectory of the aircraft is shown in Figure 3:

Assume that the basic simulation conditions are: The random drift of SINS gyro is 0.5°/h. The random offset of accelerometer is 50 μg; Initial misalignment angle is $\left[\begin{array}{ccc} 10^{\prime \prime } & 60^{\prime \prime } & 10^{\prime \prime } \end{array}\right]$. Initial state noise covariance estimation is unbiased, which is $Q\;=\;diag[w_{g}^{2},w_{g}^{2},w_{g}^{2},w_{a}^{2},w_{a}^{2},w_{a}^{2}]$, and wg=0.5π0.5π1801800.5π0.5π18018036003600, $w_{a}\;=\;50\cdot 10^{-6}g$, where *g* is the acceleration of gravity; the initial position of the aircraft is $116^{{}^{\circ}}$ of east longitude, $39^{{}^{\circ}}$ of north latitude; the shooting angle is $90^{{}^{\circ}}$; the thrust acceleration is 40 m/s${}^{2}$ at the first 60 s; in the launch inertial system, the initial pitching angle is $90^{{}^{\circ}}$ and remains the same during the first 10 s, then it changes from $90^{{}^{\circ}}$ to $30^{{}^{\circ}}$ in the form of quadratic function during the next 50 s, and then it remains the same during the rest of the time; in addition, the heading angle and rolling angle are both $0^{{}^{\circ}}$ throughout the whole process; the simulation time is 1110 s, the sampling interval is 0.01 s, and 50 Monte Carlo simulations are performed.

(1) Gaussian state noise and the estimation are unbiased:

The condition setting with Gaussian state noise and unbiased estimation is the same as the basic simulation conditions above. Taking the average of the errors, the improved federated Sage–Husa adaptive filter, federated Sage–Husa adaptive filter and the federated filter are used in integrated navigation, and the simulation error curves are shown in the Figure 4, Figure 5 and Figure 6:

It can be seen from the Figure 4, Figure 5 and Figure 6 that there are almost no differences in the navigation errors of the three methods in the three directions. The following table is a quantitative analysis.

It can be seen from the Table 1 and Table 2 that the navigation errors of the three methods in three directions are almost the same, and the subtle differences are too small to be noticed, that is, when the state noise is Gaussian and the estimation is unbiased, the three methods are roughly the same.

(2) Gaussian state noise and the estimation are biased:

The settings of the parameters are same as those in Tabel (1), and the initial estimation of state error covariance is Q=QQ1010.

It can be seen from the above Figure 7, Figure 8 and Figure 9 and the Table 3 and Table 4 that, when the estimation of the state noise is deviated, even if the state noise is Gaussian, the filter effects of the three methods are different. In the comparison of position and velocity errors, the improved federated adaptive filtering is the best, followed by the federated adaptive filter, the federated filter is not effective because it depends on the initial value of the state noise.

(3) Non-Gaussian state noise and the estimation are biased:

In the test (2), the setting of the parameters is added as follows: The SINS gyro random constant drift is 0.2°/h, the accelerometer’s random offset is 50 μg, and the initial misalignment angle is $\left[\begin{array}{ccc} 10^{\prime \prime } & 60^{\prime \prime } & 10^{\prime \prime } \end{array}\right]$. The simulation error curve is shown in the Figure 10, Figure 11 and Figure 12:

The above simulation is performed under the condition that the state noise is non-Gaussian and the estimation is biased, the tables are obtained in the case of using the federated Kalman filter, the federated adaptive filter and the improved federated adaptive filter to compare speed with position error in three directions. It can be seen from Figure 10, Figure 11 and Figure 12 that, in the initial time, the three methods have large fluctuations owing to too few samples. As the number of samples increases, the three methods get stable gradually. In addition, when the number of samples increases to a certain extent, the advantages of improved federated adaptive filter gradually appear, which is the best among the three methods, while the federated adaptive method is the second, and the federated Kalman filter is the worst. The error statistics in three directions are shown in Table 5 and Table 6.

It can be seen from the comparison of the position and velocity errors that in the integrated navigation process, the effect of the improved federated adaptive filter is better than the other two methods in the three directions.

(4) Time-varying state noise and the estimation are biased:

Let the constant offset of the gyroscope in test (3) be set to 0, and it increases to 0.2°/h with time. The random offset of the accelerometer is set to 0 at the beginning, and it evenly increases to 50 μg with time. The comparison of the three methods in three directions is as Figure 13, Figure 14 and Figure 15:

It can be seen from the above Figure 13, Figure 14 and Figure 15 and the Table 7 and Table 8 that, when the state noise is time-varying, the filter effect of the three methods is similar to the case of the non-Gaussian state noise. Improved federated adaptive filter has the best effect of the position and velocity error, followed by federated adaptive filter, while the federated Kalman filter is the worst.

Comparing the improved federated adaptive filter and federated filter in different situations, comparison of position error under the conditions of test (1) and test (4) can be taken as an example, and the precision changes of the two filters in E-N-U directions are shown in Table 9.

“+” means the precision is improved, while “−” means the accuracy is reduced. It can be seen from Table 9 that the precision of the improved federated adaptive filter has few changes for different conditions of state noise, while the federated filter’s precision decreases significantly, which shows that the improved federated adaptive filter has little dependence on the initial noise estimation, but the federated filter depends more.

Therefore, to sum up the above four cases, it can be seen that the improved federated adaptive filter algorithm can perform operations based on the state noise with unknown characteristics, and its filter accuracy is higher than the other two methods. However, since the filter algorithm designed in this paper improves the estimation of the statistical characteristics of the unknown state noise, the difference of the velocity error between the three methods is not as obvious as the position error, and it is related to the system characteristics. In summary, for different systems, the weighting mode and weighting function should be selected according to the characteristics of the system to obtain the optimal result of the federated adaptive filter.

## 7. Conclusions

In this paper, a filter algorithm based on the federated filter and simplified Sage–Husa adaptive filter is proposed for systems with time-varying state noise and biased estimation. The algorithm uses federated filter as the framework of the multi-source integrated navigation, and the local filters choose the improved Sage–Husa adaptive filter as the algorithm. In the updating process of the parameters, the federated and the adaptive principle are combined, and the exponential function is used to characterize the weighting value changes of the two updating principles, so as to obtain an improved federated adaptive algorithm with dynamic adaptive ability. Through the theoretical analysis and simulations of the improved federated adaptive algorithm, it can be seen that, when the number of samples is sufficient, the filter will tend to be stable and convergent. Compared with the federated Kalman filter and the common federated adaptive filter, the accuracy of this improved method is the highest. It shows that the improved federated Sage–Husa adaptive filter is effective in improving the federated algorithm, and it can weaken the influence of the initial estimation error of the state noise to some extent and improve the navigation accuracy.

## Figures and Tables

**Figure 1 sensors-19-03812-f001:**
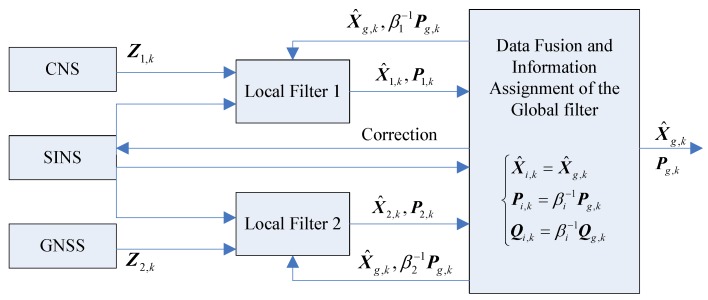
Federated filter structure of SINS/CNS/GNSS integrated navigation.

**Figure 2 sensors-19-03812-f002:**
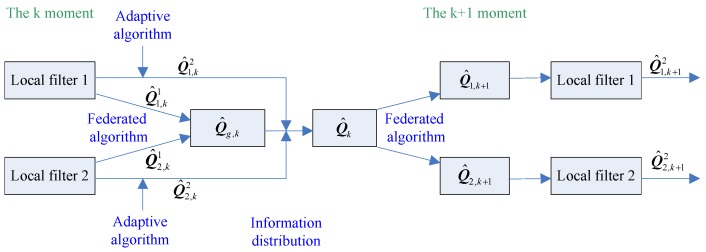
The algorithm flow of the federated adaptive filter.

**Figure 3 sensors-19-03812-f003:**
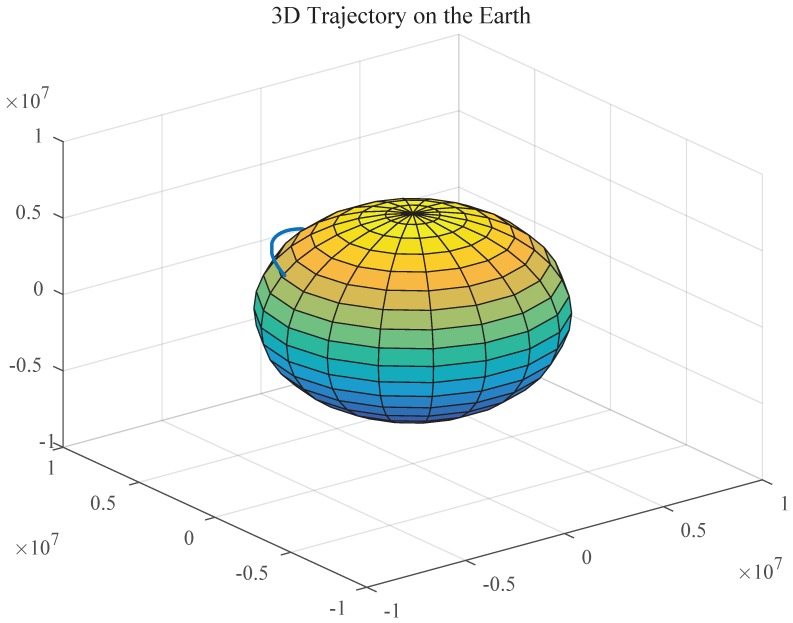
The trajectory of the aircraft on the earth.

**Figure 4 sensors-19-03812-f004:**
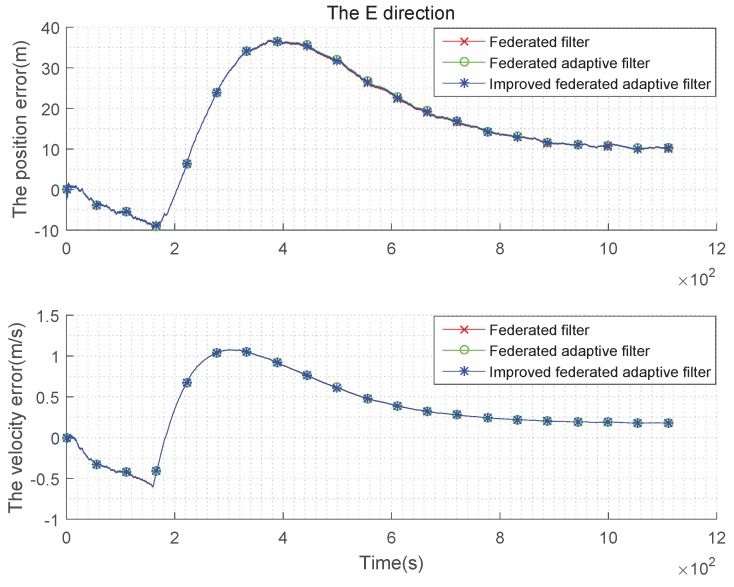
Comparison of the navigation error with Gaussian state noise and unbiased estimation in E direction.

**Figure 5 sensors-19-03812-f005:**
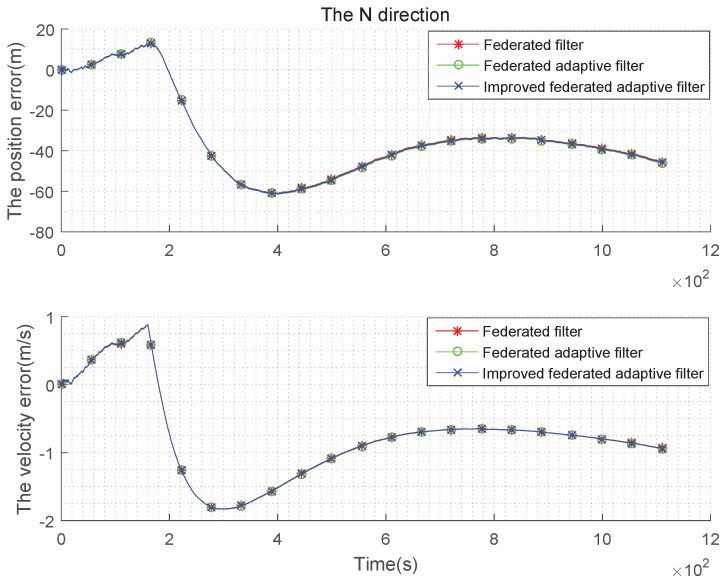
Comparison of the navigation error with Gaussian state noise and unbiased estimation in N direction.

**Figure 6 sensors-19-03812-f006:**
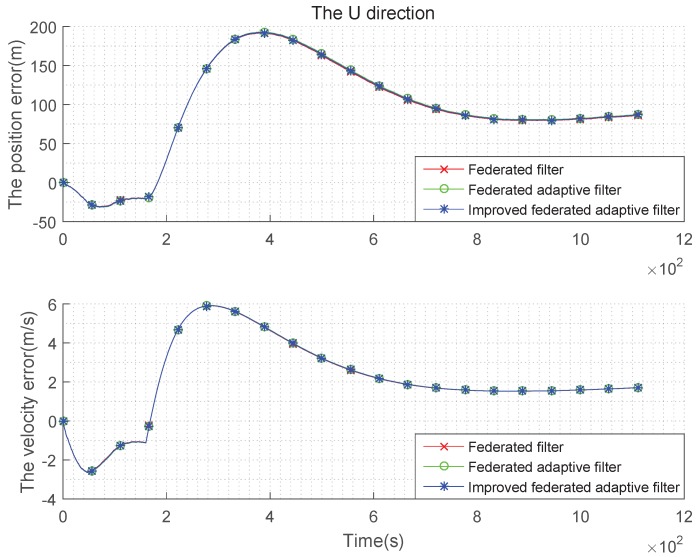
Comparison of the navigation error with Gaussian state noise and unbiased estimation in U direction.

**Figure 7 sensors-19-03812-f007:**
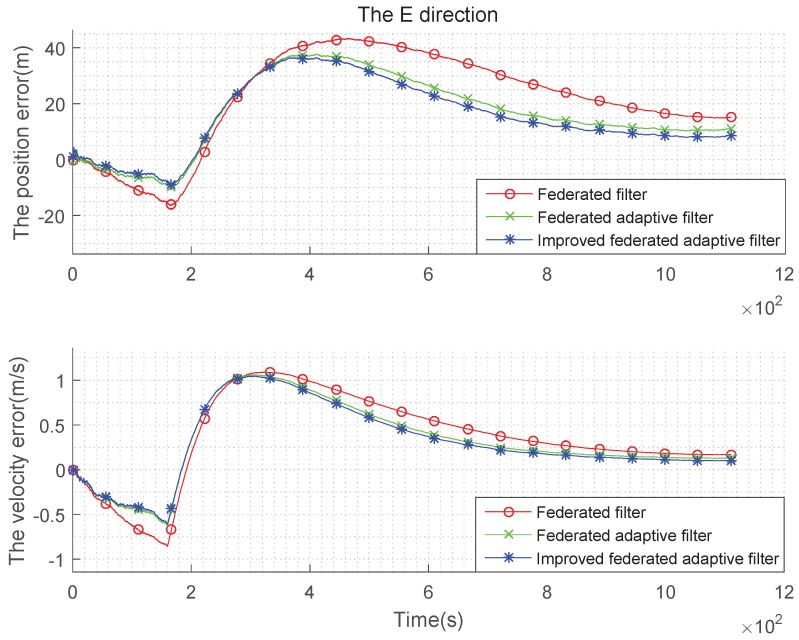
Comparison of the navigation error with Gaussian state noise and biased estimation in E direction.

**Figure 8 sensors-19-03812-f008:**
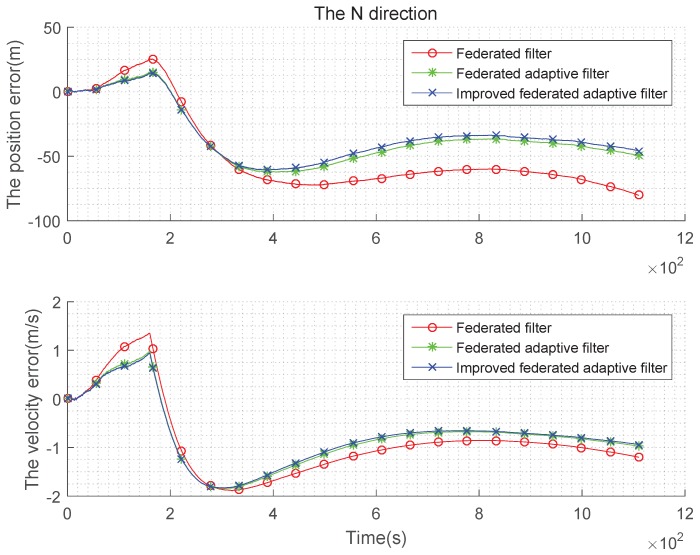
Comparison of the navigation error with Gaussian state noise and biased estimation in N direction.

**Figure 9 sensors-19-03812-f009:**
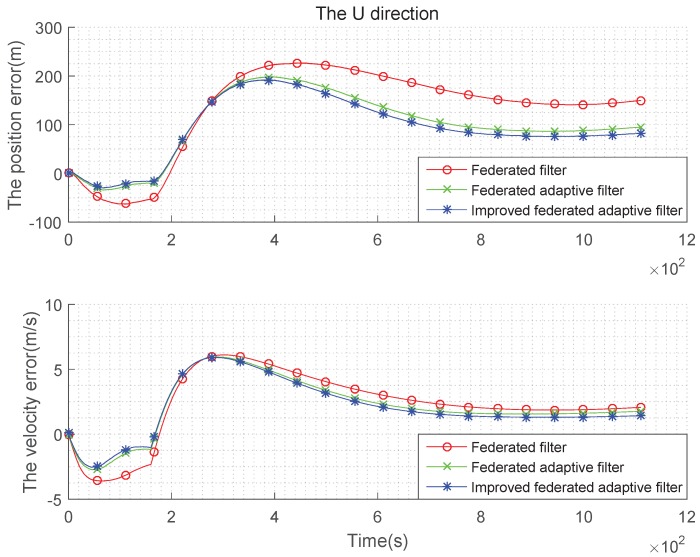
Comparison of the navigation error with Gaussian state noise and biased estimation in U direction.

**Figure 10 sensors-19-03812-f010:**
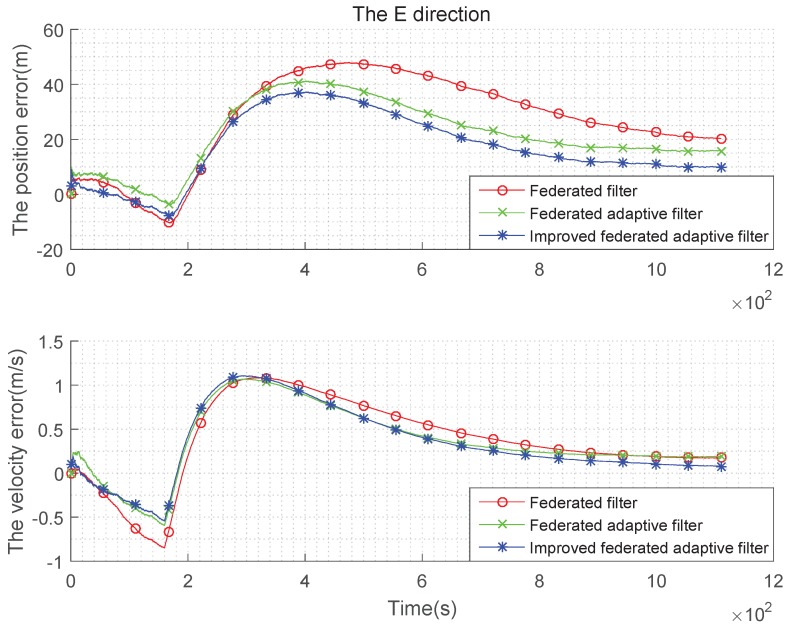
Comparison of the navigation error with non-Gaussian state noise and biased estimation in E direction.

**Figure 11 sensors-19-03812-f011:**
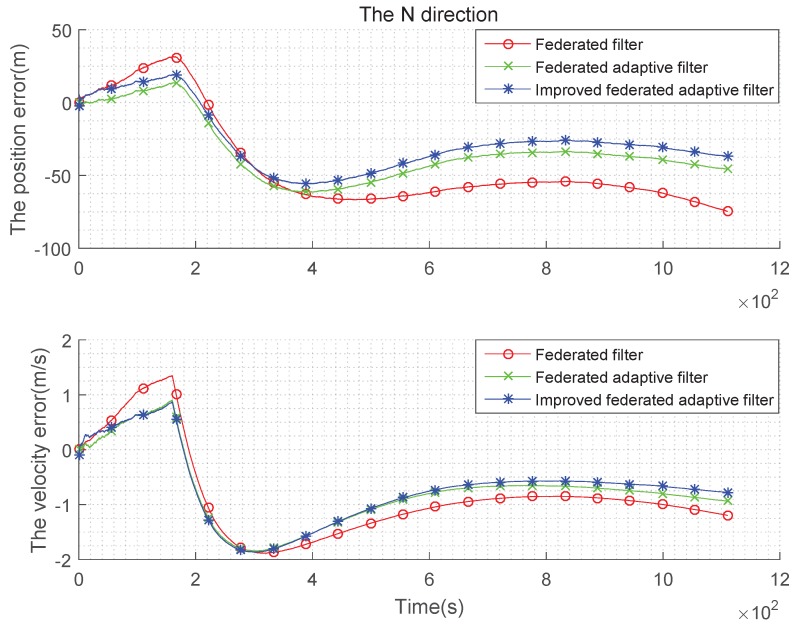
Comparison of the navigation error with non-Gaussian state noise and biased estimation in N direction.

**Figure 12 sensors-19-03812-f012:**
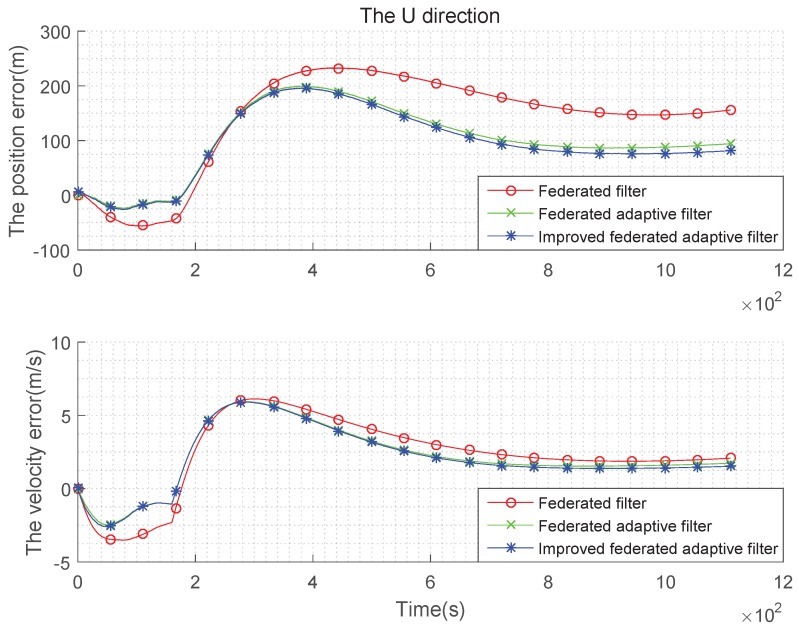
Comparison of the navigation error with non-Gaussian state noise and biased estimation in U direction.

**Figure 13 sensors-19-03812-f013:**
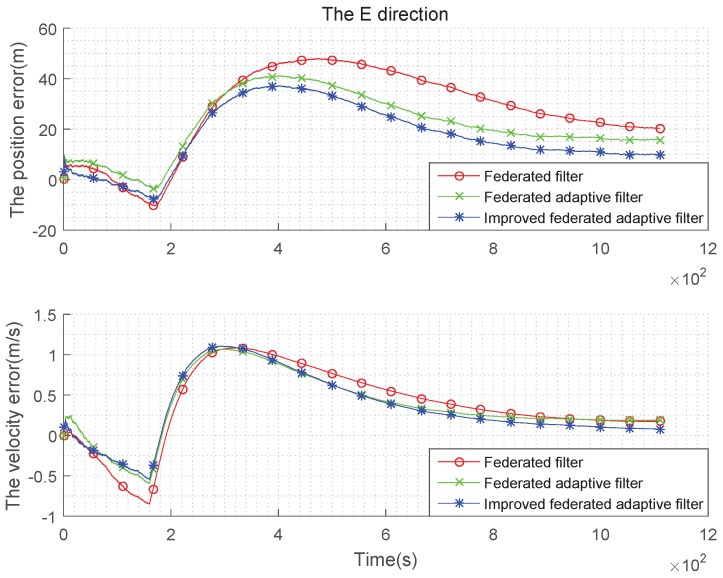
Comparison of the navigation error with time-varying state noise and biased estimation in E direction.

**Figure 14 sensors-19-03812-f014:**
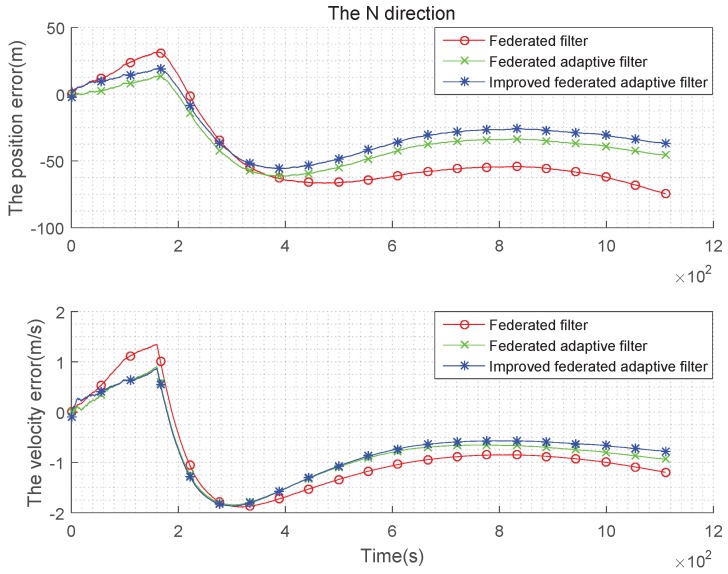
Comparison of the navigation error with time-varying state noise and biased estimation in N direction.

**Figure 15 sensors-19-03812-f015:**
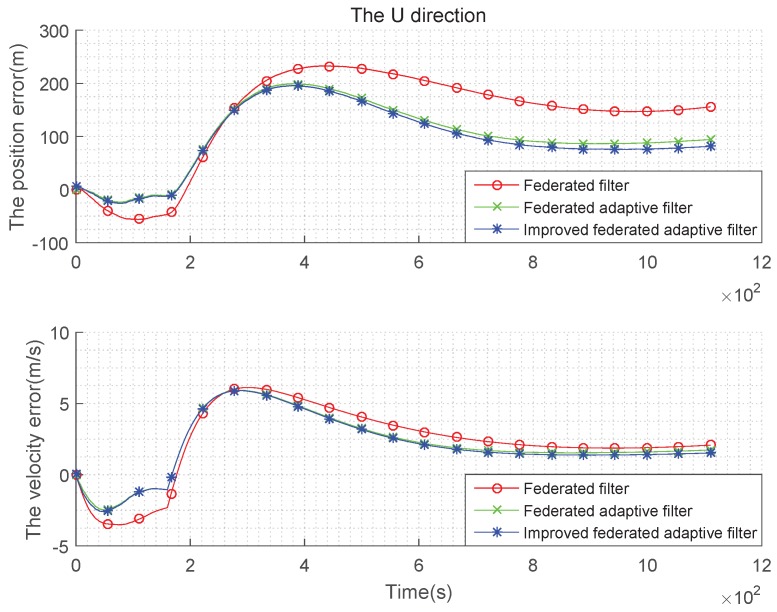
Comparison of the navigation error with time-varying state noise and biased estimation in U direction.

**Table 1 sensors-19-03812-t001:** Position error (m).

	Improved Federated Sage–Husa Adaptive Filter	Federated Sage–Husa Adaptive Filter	Federated Filter
E	17.3584	17.4424	17.2760
N	35.9106	36.1043	35.7211
U	100.0971	100.6206	99.5849

**Table 2 sensors-19-03812-t002:** Velocity error (m/s).

	Improved Sederated Sage–Husa Adaptive Filter	Federated Sage–Husa Adaptive Filter	Federated Filter
E	0.4472	0.4481	0.4463
N	0.9144	0.9166	0.9122
U	2.5977	2.6043	2.5912

**Table 3 sensors-19-03812-t003:** Position error (m).

	Improved Federated Sage–Husa Adaptive Filter	Federated Sage–Husa Adaptive Filter	Federated Filter
E	16.3954	18.1059	24.3230
N	35.9874	38.1979	52.8275
U	98.6435	107.0620	147.8313

**Table 4 sensors-19-03812-t004:** Velocity error (m/s).

	Improved Federated Sage–Husa Adaptive Filter	Federated Sage–Husa Adaptive Filter	Federated Filter
E	0.4073	0.4316	0.5174
N	0.9145	0.9399	1.0951
U	2.4948	2.6905	3.1695

**Table 5 sensors-19-03812-t005:** Position error (m).

	Improved Federated Sage–Husa Adaptive Filter	Federated Sage–Husa Adaptive Filter	Federated Filter
E	17.8912	21.8945	28.2643
N	31.3211	35.8689	49.4581
U	99.0952	104.7118	151.0506

**Table 6 sensors-19-03812-t006:** Velocity error (m/s).

	Improved Federated Sage–Husa Adaptive Filter	Federated Sage–Husa Adaptive Filter	Federated Filter
E	0.4204	0.4456	0.5107
N	0.8720	0.9126	1.1015
U	2.5280	2.6031	3.1570

**Table 7 sensors-19-03812-t007:** Position error (m).

	Improved Federated Sage–Husa Adaptive Filter	Federated Sage–Husa Adaptive Filter	Federated Filter
E	18.2575	19.7800	26.8285
N	34.3613	36.0846	50.2126
U	99.6093	103.6305	150.2649

**Table 8 sensors-19-03812-t008:** Velocity error (m/s).

	Improved Federated Sage–Husa Adaptive Filter	Federated Sage–Husa Adaptive Filter	Federated Filter
E	0.4225	0.4541	0.5248
N	0.8606	0.9161	1.0905
U	2.5309	2.6213	3.1736

**Table 9 sensors-19-03812-t009:** The precision changes of the two filters in E-N-U directions.

	Improved Federated Sage–Husa Adaptive Filter	Federated Filter
E	−5.1796%	−55.2934%
N	4.5089%	−40.5685%
U	0.4897%	−50.8912%

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
