# Peer review of "SINS/CNS/GNSS Integrated Navigation Based on an Improved Federated Sage–Husa Adaptive Filter"

_sensors, 2019, doi:10.3390/s19173812_

Round 1
Reviewer 1 Report
1. What does the "main filter" in equation (1) mean?
You have assumed that there is no main filter in the simulations.
You should describe How the main filter differs from the local filters.
2. You seem to be using FR mode in FR(Fusion-Reset) mode and FT(Fault-Tolerance) mode,
I recommend that you mention why you chose FR mode design.
3. When designing a federated filter,
it is not a common practice to distribute the process noise covariance information of the
master filter(=global filter) to the local filters.
You should describe why you designed it.
4. Ref [15] describes that the MAP problem of multichannel Bernoulli-Gaussian input sequences
can be separated into MAP detection and MV estimation problem.
The reference does not contain what you mentioned.
5. In equation (26), n means N? Why do you treating P_m differently than P_i?
6. What does the "F norm" in equation (27) mean?
7. In section 4.2.1, you mentioned that "The traditional federated Kalman filter does not have the ability to eliminate the influence of deviation".
You have to make it clear why you mentined it.
8. Why did you assume that "var(Q_i,k+1^1)=0" and "bias(Q_i,k+1^2)=0"?
9. Why do you have to satisfy the following conditions "var(Q_i,k+1^2) < var(Q_i,k^2)?
10. If you use the INS/GNSS integrated navigation as the local filter for the proposed federated filter, the navigation errors shown in the simulations is too large.
Describe the GNSS and Celestial navigation error model specification.
11. The sensor noise of INS is largely divided into the deterministic noise
and stochastic (random) noise.
In general, the deterministic noise such as bias, scale factor, and misalignment noise can be eliminated using proper calibration method.
Why do you consider the biased state noise?
12. Designs that combine INS/GNSS and INS/CNS will show novelty,
However, you have to describe the differences between existing SHAKF and the proposed SHAKF.
and between existing Federated filter and the proposed Federated filter.
Author Response
The authors thank the reviewer for the encouragement for this article. As for the improvements, we have made the following changes:
Point 1: What does the "main filter" in equation (1) mean? You have assumed that there is no main filter in the simulations. You should describe How the main filter differs from the local filters.
Response 1: Since the traditional federated filter is a distributed filter with a two-stage structure, the main filter is used to fuse with each sub-filter to complete the final global optimal estimation. The reason that the main filter is not selected in this paper is that the sub-filter has time update and measurement update process, while the main filter only has a time update process, the accuracy of the global optimal estimation will be decreased. In eq. (1), since the main filter has no measurement process, the value range of i is different in the two equations. In addition to this, the main filter is optional, so it needs to be subscripted. And I will correct it in the revision process.
Point 2: You seem to be using FR mode in FR(Fusion-Reset) mode and FT(Fault-Tolerance) mode, I recommend that you mention why you chose FR mode design.
Response 2: For traditional No-reset fault-tolerant federated filter, the main filter does not reset the local filter information. But in order to meet the adaptive requirements of the system, a feedback process needs to be performed after each step of the filter algorithm is completed. Therefore, the improved federated filter reset method proposed in this paper uses the eq.(2) and (4) to complete the information fusion process through the allocation and addition of global filter and local filter without the participation of the main filter.The dynamic trend of the system will be included in the feedback parameter changes, and through this process, the proportion of some operators can be adjusted. This structure is actually a FR(Fusion-Reset) mode, and it is more accurate without considering the failure of the local filter. These explanations will be added in the chapter 2.
Point 3: When designing a federated filter, it is not a common practice to distribute the process noise covariance information of the master filter(=global filter) to the local filters. You should describe why you designed it.
Response 3: In the No-Reset mode of federated filter, the global filter has no information distribution to the subfilter. However, the design principle of the federated filter in this paper is to continuously correct the distribution factor through the allocation and integration process of the global filter and the local filter, so as to achieve the adaptive requirements of the system. Therefore, this paper has the allocation and integration of the process noise covariance, and I will add the design ideas in the revisions.
Point 4: Ref [15] describes that the MAP problem of multichannel Bernoulli-Gaussian input sequences can be separated into MAP detection and MV estimation problem. The reference does not contain what you mentioned.
Response 4: Ref [15] is not appropriate, I will replace it with a more appropriateone.
Point 5: In equation (26), n means N? Why do you treating P_m differently than P_i?
Response 5: In the expression of Eq.(26), n is the same as N in the previous text, and I am sorry here that the expression is not uniform, and I will modify it. The reason for the different expressions of P_n and P_m is that not all federated filters have a main filter, so the two are not written together for more rigor.
Point 6: What does the "F norm" in equation (27) mean?
Response 6: The reason of using F norm of P is the same as using the trace of P , and the distribution factors are determined according to the proportion of the process noise covariance of each sub-filter. In addition to the F-norm method, there are other calculation methods, but this paper only lists two common ones here.
Point 7: In section 4.2.1, you mentioned that "The traditional federated Kalman filter does not have the ability to eliminate the influence of deviation". You have to make it clear why you mentined it.
Response 7: The meaning of this sentence in the text is that if the deviation is existed when set the initial value, the traditional federated filter cannot eliminate the deviation in the subsequent calculation, so the accuracy of the filter will continue to go down. For this reason, an adaptive algorithm is needed to solve the influence of the deviation. Therefore, this paper combines the federated algorithm and the adaptive algorithm to obtain a federated adaptive filter with certain adaptive capabilities.
Point 8: Why did you assume that "var(Q_i,k+1^1)=0" and "bias(Q_i,k+1^2)=0"?
Response 8: In the combination process of federated algorithm and adaptive algorithm, both algorithms have advantages and disadvantages. For example, the initial estimation deviation of the federated algorithm will always exist. When there are too few samples, there will be large fluctuations of the adaptive algorithm. Therefore, at the initial stage, the federated algorithm process error mainly comes from the deviation while the adaptive algorithm process error mainly comes from the variance, and the combined weight of the two algorithms needs to be changed continuously. The change is reflected on the selection of the proportional coefficient, which is also the core of the federated adaptive algorithm. In order to obtain the trend of the proportional coefficient, it is necessary to set the initial value of the two algorithms. For the sake of convenience, according to the variation characteristics of the weight, the initial variance of (Q_i,k+1^1) in the federated algorithm is set to 0, and the initial deviation of (Q_i,k+1^2) in the adaptive algorithm is set to 0.
Point 9: Why do you have to satisfy the following conditions "var(Q_i,k+1^2) < var(Q_i,k^2)?
Response 9: The meaning of this inequality is that, the adaptive algorithm is given, the process noise variance is continuously decreasing. This assumption is based on the operational characteristics of adaptive filter. Since there are too few number of samples at the beginning, the characteristics of the system obtained by adaptive filter are also less. Therefore, the estimation of the variance of the state noise is not accurate enough, and the fluctuation is large, that is, its variance is larger; When the filter is becoming steady, the estimation of the variance of the state noise has stabilized, and its variance is small. Therefore, based on this feature, this inequality is set.
Point 10: If you use the INS/GNSS integrated navigation as the local filter for the proposed federated filter, the navigation errors shown in the simulations is too large. Describe the GNSS and Celestial navigation error model specification.
Response 10: The sampling frequency of INS is much higher than that of GNSS. The error of integrated navigation mainly comes from the error accumulation of INS. In this paper, the error of INS is not eliminated in order to verify the feasibility of the improved algorithm. Therefore, this error is reflected in the simulation process.
This paper mainly discusses the advantages of the federated adaptive filter, and the data in the simulation part is simulated based on the ballistic data. The main purpose is to verify the effectiveness of the proposed method. Regarding the error specification of GNSS and CNS, I will supplement and improve the simulation based on the data source and experimental conditions.
Point 11: The sensor noise of INS is largely divided into the deterministic noise and stochastic (random) noise. In general, the deterministic noise such as bias, scale factor, and misalignment noise can be eliminated using proper calibration method. Why do you consider the biased state noise?
Response 11: Since this paper is to verify the validity of this federated adaptive algorithm, it is necessary to retain the necessary error to determine whether this method can accomplish better target localization under various conditions.
Point 12: Designs that combine INS/GNSS and INS/CNS will show novelty, However, you have to describe the differences between existing SHAKF and the proposed SHAKF. and between existing Federated filter and the proposed Federated filter.
Response 12: For the improved federated adaptive filter algorithm proposed in this paper, the problem of slow convergence speed and low precision estimation in the traditional federated adaptive method is mainly solved. The advantage of the method is proved in the simulation. For the comparison of the two algorithms in theory and simulation, the article will be supplemented and improved, thank you for the reminders.
Finally, I would like to thank the reviewer for your pertinent comments, and I will continue to improve this manuscript.

Reviewer 2 Report
This paper provides the design of an improved federated Sage-Husa adaptive filter, which integrates the measurements of SINS/CNS/GNSS. The main purpose of this paper is to achieve better filter performance, supposing the system has time-varying noise or mis-estimated state noise. The designed filter structure federates two local filters, i.e. SINS/CNS and SINS/GNSS filters, and a global filter integrating the result of the two local filters. Each local filter applies a simplified Sage-Husa adaptive filter. The main contribution of this work is proposing a weighting method, which fuses the federated and adaptive update of the state noise covariance. The designed filter is validated by a simulation. The experimental result is evaluated, with respect of the accuracy improvement of the positioning and velocity solution, with or without applying the improved filter structure.
Broad comments :
The consistency of abbreviation explanations should be maintained: MSE is explained in the main context, while SINS/CNS/GNSS are not explained. Moreover, there is no hint that it exists a list of abbreviations after the main context. The reviewer suggests using bold lowercase letters to represent vectors, in order to distinguish vectors from matrices, e.g. system state is a vector instead of a matrix. Please consider defining and describing every used symbol clearly, e.g. qk and rk are left undefined and unexplained. Please consider keeping the consistency of the notations and try to give an overall definition of the notation, e.g. ∧ for predicted value of a defined variable. Please consider explaining the equations, e.g. Equation (16) ~ (22) are left without any explanation. If the author intends to leave them without explanation, please consider putting these equations into an appendix. The measurement setup for the experimental validation is not clear. Please try to answer the following questions in the main context. What does it mean by using ‘simulation’? Does it mean post-processing of recorded data from real sensors or simulation of all used sensors? If it is the second case, then, how is the INS/CNS/GNSS data simulated, with which software/hardware? What kind of GNSS signal is simulated and used? Which system? Which frequency? Which kind of observables, pseudorange or carrier phase? Because these questions are relevant to understand the navigation accuracy in the experimental results. In the experimental results, the designed filter are evaluated with respect of the navigation accuracy. Which reference system is used to calculate the position and velocity error? If we take the simulation setting 1), which is the perfect working condition for the filter, as the baseline. In other settings 2), 3) and 4), the filter works with worse condition then the baseline, therefore, it is expected that with other settings the filter shall have worse performance then the baseline setting. This is observed by federated filter and federated adaptive filter. However, this is not the case by improved federated adaptive filter. The improved federated adaptive filter has a better performance with worse condition. Would the author discuss this result? Please consider other indexes to evaluate the result. Due to the long simulation time, the standard deviation might also be an interesting index to evaluate the filter performance. The figures are high quality vector graphics. However, the illustration might be improved by using solid lines with points, instead of using dashed lines.
Specific comments:
Line 59: Sage-Husa adaptive filter is chosen as… Line 59: ‘as the sub-filters’ algorithm’ could be simplified as ‘as the sub-filter’ Line 114: Qg,0 are given? What does it mean by ‘given’. Are these initial values of state covariance matrice predefined parameters? The line under line 130: what does it mean by ‘the noise Wk is inaccurate’? How can noise be inaccurate? Line 143: ‘At this time’ --> ‘In this case’ Line 257: ‘Table 12’ --> ‘Table 1&2’ or ‘Table 1 and Table 2’ Line 258: ‘too small to be ignored’ --> ‘too small to be noticed’ The axis title of figures should be left aligned All x-axes (time axes) are the same. Please combine some of them to save the space. Please consider to rearrange the place of the time scale along the x-axis to save the space.
Author Response
Thanks to the reviewer for his meticulous comments, I think these are very important for the improvement of my manuscript.
Ponit 1: The consistency of abbreviation explanations should be maintained: MSE is explained in the main context, while SINS/CNS/GNSS are not explained. Moreover, there is no hint that it exists a list of abbreviations after the main context. The reviewer suggests using bold lowercase letters to represent vectors, in order to distinguish vectors from matrices, e.g. system state is a vector instead of a matrix. Please consider defining and describing every used symbol clearly, e.g. qk and rk are left undefined and unexplained. Please consider keeping the consistency of the notations and try to give an overall definition of the notation, e.g. ∧ for predicted value of a defined variable. Please consider explaining the equations, e.g. Equation (16) ~ (22) are left without any explanation. If the author intends to leave them without explanation, please consider putting these equations into an appendix. The measurement setup for the experimental validation is not clear. Please try to answer the following questions in the main context. What does it mean by using ‘simulation’? Does it mean post-processing of recorded data from real sensors or simulation of all used sensors? If it is the second case, then, how is the INS/CNS/GNSS data simulated, with which software/hardware? What kind of GNSS signal is simulated and used? Which system? Which frequency? Which kind of observables, pseudorange or carrier phase? Because these questions are relevant to understand the navigation accuracy in the experimental results. In the experimental results, the designed filter are evaluated with respect of the navigation accuracy. Which reference system is used to calculate the position and velocity error? If we take the simulation setting 1), which is the perfect working condition for the filter, as the baseline. In other settings 2), 3) and 4), the filter works with worse condition then the baseline, therefore, it is expected that with other settings the filter shall have worse performance then the baseline setting. This is observed by federated filter and federated adaptive filter. However, this is not the case by improved federated adaptive filter. The improved federated adaptive filter has a better performance with worse condition. Would the author discuss this result? Please consider other indexes to evaluate the result. Due to the long simulation time, the standard deviation might also be an interesting index to evaluate the filter performance. The figures are high quality vector graphics. However, the illustration might be improved by using solid lines with points, instead of using dashed lines.
Response 1: According to the reviewer's suggestions, the author added an explanation of the abbreviations in the text, and summarized all the abbreviations at the end of the paper for easy reading. The author explains qk and rk in the equations (16)~(22), reading the whole paper and adding missing explanations.
This paper mainly discusses the advantages of improved federated adaptive filter, mainly used in theoretical research. Therefore, for the simulation part, there is no actual measured data , but the relevant data of the aircraft is simulated by software simulation, which is derived from missile ballistic data. The author will add corresponding explanations in the simulation part.
The improvement of the federated adaptive filter with better accuracy under poor conditions can be explained. The main reasons are as follows: The main application background of the improved federated adaptive filter proposed in this paper is to estimate the existence of bias and time-varying noise. As the number of samples increases, the accuracy of adaptive estimation will also increase. When the initial estimation is accurate, that is, the condition is set as 1), the improved federated adaptive filter still obtains the estimated values from the samples, so the estimation is still inaccurate. Therefore, under the four setting conditions in the simulation, the improved adaptive federated filter proposed in this paper has similar accuracy, but the noise has a certain randomness, so it is common that a slightly better result is obtained in poor conditions.
This paper mainly proves the advantages of the proposed improved federated adaptive filter method, so it does not eliminate the constant error of gyroscope and accelerometer. However, the method can still accomplish the target localization better without estimating the constant error, which proves the advantages of the method.
Ponit 2: Line 59: Sage-Husa adaptive filter is chosen as… Line 59: ‘as the sub-filters’ algorithm’ could be simplified as ‘as the sub-filter’ Line 114: Qg,0 are given? What does it mean by ‘given’. Are these initial values of state covariance matrice predefined parameters? The line under line 130: what does it mean by ‘the noise Wk is inaccurate’? How can noise be inaccurate? Line 143: ‘At this time’ --> ‘In this case’ Line 257: ‘Table 12’ --> ‘Table 1&2’ or ‘Table 1 and Table 2’ Line 258: ‘too small to be ignored’ --> ‘too small to be noticed’ The axis title of figures should be left aligned All x-axes (time axes) are the same. Please combine some of them to save the space. Please consider to rearrange the place of the time scale along the x-axis to save the space.
Response 2: The expression “…… Qg,0 are given” is not appropriate, I will change it to “…… Qg,0 are known”in the text. And initial values of state covariance matrice are predefined parameters. As to “the noise Wk is inaccurate”, in practice, the defined state equation does not necessarily satisfy the actual operating state, and the state error variance is inaccurate, and there is a certain discrepancy with the estimation.
Finally, I would like to thank the reviewer for your pertinent comments, and I will continue to improve this manuscript.

Round 2
Reviewer 1 Report
The authors answered all the comments sincerely.
I also think that the modified paper logically describes what it intended.
Therefore, I agree to publish this paper.
Reviewer 2 Report
The reviewer appreciate that the authors have considered the suggestions from the reviewer and adjusted the manuscript according to the review points.
As to the experimental validation, the explanation of the software simulation and the accuracy improvement is adequate. Still, the reviewer strongly suggest using the real data or real hardware to validate the filter design in the future work, to prove the robustness of the filter against the real world measurement noise, or even sensor failure. Further, it is easier to understand the accuracy or accuracy improvement, if real GNSS system is used.